# Mitofilin–mtDNA Axis Mediates Chronic Lead Exposure-Induced Synaptic Plasticity Impairment of Hippocampal and Cognitive Deficits

**DOI:** 10.3390/biom15020272

**Published:** 2025-02-12

**Authors:** Lihong Su, Jinchao Hou, Boxuan Wang, Yuqi Li, Xiaodong Huo, Tao Wang, Yuankang Zou, Gang Zheng

**Affiliations:** 1School of Public Health, Gansu University of Chinese Medicine, Lanzhou 730000, China; 2Department of Occupational and Environmental Health and The Ministry-of-Education’s Key Laboratory of Hazard Assessment and Control in Special Operational Environment, School of Preventive Medicine, Fourth Military Medical University, 169 Changlexi Road, Xi’an 710032, China

**Keywords:** lead exposure, mitofilin, mtDNA, neuron, synaptic plasticity

## Abstract

Neurotoxic damage resulting from lead pollution exposure constitutes a significant public health concern. The regulatory impact of lead (Pb) exposure on neuronal dendritic spine plasticity, a crucial mechanism for neuronal adaptation, warrants further investigation. To elucidate the role and mechanism of the Mitofilin–mtDNA axis in hippocampal synaptic plasticity and learning and memory impairment induced by lead exposure, in this study, both in vivo and in vitro models were subjected to chronic lead exposure. The results showed that the spatial learning and memory abilities of lead-exposed mice were significantly reduced. Furthermore, Western blotting and RT-PCR analyses demonstrated a significant down-regulation in the expression of the mitochondrial inner membrane protein Mitofilin. Extended exposure to lead has the potential to compromise the plasticity of dendritic spines within the CA1 region of hippocampal neurons and disrupt the structural integrity of neuronal mitochondria. Furthermore, lead exposure was associated with elevated levels of malondialdehyde (MDA) and reactive oxygen species (ROS) in neurons. The study additionally demonstrated that the overexpression of Mitofilin ameliorated deficits in spatial learning and memory in mice subjected to chronic lead exposure. This overexpression also facilitated the normal formation of neuronal dendritic spines, preserved the structural integrity of the mitochondrial inner membrane, and mitigated mitochondrial damage. The study further revealed that the overexpression of Mitofilin markedly suppressed the release of mitochondrial DNA (mtDNA) in neurons subjected to chronic lead exposure, while concurrently reducing the expression levels of the inflammasome Nlrp3 and the inflammatory cytokine IL-1β. Additionally, there was a significant reduction in the levels of malondialdehyde (MDA) and reactive oxygen species (ROS) in lead-exposed neurons with Mitofilin overexpression. These findings suggest that the mitochondrial inner membrane protein Mitofilin may play a role in mediating synaptic plasticity impairment following chronic lead exposure through the regulation of mitochondrial function.

## 1. Introduction

Environmental exposure to lead can occur through ingestion, inhalation, dermal absorption, and transplacental transfer. Lead exposure is associated with irreversible damage to the skeletal, cardiovascular, reproductive, and renal systems, with particularly detrimental effects on the central nervous system (CNS) [1,2]. The neurotoxic effects of lead have been acknowledged as a significant public health concern in in China, France, the United States, and other countries [3]. Research has demonstrated that prolonged exposure to lead can lead to mitochondrial dysfunction and aberrant oxidative stress, culminating in synaptic plasticity impairment and neuroinflammation. Additional studies have indicated that lead exposure may contribute to the onset and progression of neurodegenerative diseases, such as Alzheimer’s disease, by disrupting synaptic plasticity and through other mechanisms in the early stages [4,5,6]. However, the precise pathogenesis and key targets require further investigation.

The hippocampus, a critical functional region of the brain, is integral to learning and memory processes [7]. Neurons serve as the fundamental functional units of the nervous system. Synaptic plasticity, which facilitates neurotransmitter release and transmission between neurons, constitutes a crucial neurobiological foundation for learning and memory [8,9,10]. Dendritic spines, which are small, specialized protrusions from neuronal dendrites predominantly situated at excitatory synapses, undergo structural and numerical alterations during synaptic plasticity. Research indicates that chronic lead exposure exerts a profound effect on the developing brain, leading to behavioral alterations, cognitive deficits, sympathetic excitation, and autonomic nervous system dysfunction, among other consequences [11,12,13,14]. The hippocampus, recognized as a sensitive functional region of the brain, is integral to learning and memory processes [8,9,10]. Additional studies have demonstrated that chronic lead exposure can impair synaptic plasticity within the hippocampus of the developing brain [15,16,17].

Mitochondria are unique among mammalian cell organelles in exerting genetic influence beyond the confines of the nucleus [18]. Synaptic plasticity in the adult brain is believed to represent the cellular mechanisms of learning and memory. Mitochondria are involved in the regulation of the complex processes of synaptic plasticity [19]. Mitofilin, a protein localized to the inner mitochondrial membrane, constitutes a critical component of the mitochondrial inner membrane organizing system (MINOS) complex. This protein is essential for preserving the structural integrity and functional efficacy of the mitochondrial cristae [20,21,22]. Research has demonstrated that aberrant expression of the Mitofilin protein is associated with neurodegenerative conditions, including Parkinson’s disease, as well as cardiovascular and other diseases [23,24]. Nonetheless, the involvement of Mitofilin in mediating the effects of lead exposure on synaptic plasticity remains unclear. To address this gap, the present study employed both in vitro and in vivo experimental approaches to comprehensively investigate the role and underlying mechanisms of Mitofilin in synaptic plasticity damage induced by chronic lead exposure.

## 2. Methods and Materials

### 2.1. Materials

Lead acetate (cat. no. 316512) and sodium 2% pentobarbital (cat. no. S564508) were purchased from Merck KGaA (ShangHai, China) Fetal bovine serum (cat. no. 13011-8611) was purchased from Tianhang Biotechnology Co., Ltd. (Huzhou, China). DMEM medium (cat. no. C11995500BT), Trypsin-EDTA (cat. no. 25200072), and cyanine and streptomycin (cat. no. 15240112) were purchased from Thermo Fisher Scientific Inc. (Shanghai, China). Bovine serum albumin (BSA) blocking solution (cat. no. FC0077) and Triton X-100 (cat. no. 194854) were purchased from MP Biomedicals (ShangHai, China). 4% paraformaldehyde (cat. no. BL539A) was purchased from Biosharp Company (Hefei, China). Cell culture slides (cat. no. 801007) were purchased from NEST Company (WuXi, China). Cell lysis buffer (cat. no. P0013B), phosphatase inhibitors (cat. no. PL026-5), PMSF (cat. no. C11995500BT), loading buffer, electrophoresis buffer (cat. no. P0014D), transfer buffer (cat. no. P0021B), malondialdehyde (MDA) assay kit (cat. no. S0131M), poly-L-lysine (cat. no. ST509), and DAPI anti-fluorescence quenching mounting medium (cat. no. P0131-5ml) were purchased from Beyotime Biotechnology Company (ShangHai, China). The reactive oxygen species (ROS) detection kit (cat. no. E-BC-K138-F) was purchased from Wuhan Elabscience Biotechnology Co., Ltd. (WuHan, China). The BCA protein quantification kit (cat. no. 23225) was purchased from Thermo Company (ShangHai, China). A sterile phosphate-buffered saline (PBS) (cat. no. MI00625), membrane regeneration solution (cat. no. MI00631), and ECL chemiluminescent reagent (cat. no. MI00607A) were purchased from Mishu Bio (Xi'an, China). Rabbit anti-Mitofilin antibody (cat. no. 10179-1-AP), rabbit anti-β-actin antibody (cat. no. 81115-1-RR), rabbit anti-Tom20 antibody (cat. no. 11802-1-AP), goat anti-rabbit secondary antibody (cat. no. SA00001-2), and goat anti-mouse secondary antibody (cat. no. RGAM601) were purchased from Proteintech Group, Inc (WuHan, China). Mouse anti-DNA antibody (cat. no. 61014) was purchased from Progen (Heidelberg, Germany). Fluorescent Rabbit red secondary antibody (cat. no. 8889S) and fluorescent mouse green secondary antibody (cat. no. 4408S) were purchased from CST (Shanghai, China). Isopropyl alcohol (cat. no. 67-63-0) and methanol (cat. no. 67-56-1) were purchased from Fuyu Chemical Company (TianJin, China). DEPC Water (cat. no. R1600) was purchased from Solarbio (Beijing, China). The reverse transcription kit and real-time fluorescent quantitative PCR test kit (cat. no. RR820A) were purchased from TAKARA (Beijing, China). Immt primers (F–Immt: AGGGAGACACTCCAGCTTCA, R-Immt: CTTGCTTTTCCTGTTGCGCT) and β-actin primers (F-actin: TCATCACTATTGGCAACGAGC; R-actin: AACAGTCCGCCTAGAAGCAC) are synthesized in Sangon Biotech. The water maze, open field test, and novel object recognition test system were purchased from Xinruan Company (Shanghai, China). The Forma Series II CO_2_ cell culture incubator was purchased from Thermo Fisher Company (Shanghai, China). The graphite furnace atomic absorption spectrophotometer was purchased from Perkin Elmer Company (Shanghai, China). The Infinite F200 microplate reader was purchased from Tecan Company (Shanghai, China). The Mini Protean protein electrophoresis system was purchased from Bio-Rad Company (Shanghai, China). The ultra-high-resolution live cell imaging system was purchased from Zeiss Company (Shanghai, China).

### 2.2. Animals and Cell

The HT22 mouse hippocampal neuron cell line, obtained from Guangzhou Genio Biotechnology Co., Ltd. (Guangzhou, China), was utilized for the initial three generations in this study. Additionally, thirty-two male C57BL/6 mice, aged three weeks, were supplied by the Animal Experimental Center of the Air Force Military Medical University. The experimental protocol received ethical approval from the Experimental Animal Center of the Air Force Military Medical University, as indicated by License No: SCXK (Shaanxi) 2024-002(26 February 2019).

### 2.3. Cell Culture and Exposure to Pb

HT22 cells were cultured in DMEM medium containing 10% fetal bovine serum and 1% penicillin–streptomycin solution at 37 °C and 5% CO_2_ saturation humidity. The cells were passed every 2d, and the cell inoculation density was 6 × 104 cells/mL. HT22 cells in the control group (Con) were cultured with DMEM medium, and HT22 cells in the lead exposure group (Pb) were cultured with 5 μmol/L lead acetate medium for 24 h.

### 2.4. Animals and Treatments [25]

C57BL/6 mice were purchased from the Experimental Animal Center of the Air Force Military Medical University (Xi'an, China), production certificate number: SCXK (Shaanxi) -2019-001. C57BL/6 male mice aged 3 weeks were reared for 5 days. The mice were randomly divided into 6 groups. There were 8 rats in each group, which were the control group (Con), Pb exposure group (Pb), control + blank treatment group (Con+LV-NC), Pb exposure + blank treatment group (Pb+LV−NC), control + over expressing Mitofilin group (Con+LV-Mitofilin), and Pb exposure + over expressing Mitofilin group (Pb + LV-Mitofilin). The anesthetized C57BL/6 mice were mounted into the stereotaxic frame. A needle was vertically and bilaterally lowered into the hippocampus (ML: ±1.5; AP: −1.9; DV: −2.0 mm). A total of 0.3 μL AAV-SYN- tdTomato-P2A-Mitofilin (LV-Mitofilin) or AAV-SYN- tdTomato-P2A (LV-NC) was then bilaterally infused. The above two viruses were synthesized by Heyuan Biotechnology Co., Ltd. (Tianjin, China). Before the experiment began, the initial weights of the mice were recorded, for the Con group (14.53 ± 1.95)g and Pb group (14.40 ± 1.86)g, and the Con+LV-NC group (14.31 ± 1.87)g, Pb+LV−NC group (14.45 ± 1.98)g, Con+LV-Mitofilin group (14.62 ± 1.72)g, and Pb+LV-Mitofilin group (14.37 ± 1.79) g. Mice in the Con group, Con+LV-NC group, and Con+LV-Mitofilin group drank high-pressure deionized water, and mice in the Pb group, Pb+LV−NC group, and Pb+LV-Mitofilin group drank high-pressure deionized water containing 100 ppm Pb [26] within three months, and with the water supplemented once a week at a fixed time to ensure that the mice drink enough water.

### 2.5. Graphite Furnace Atomic Absorption Spectrometry (AAS)

Following anesthesia of the mice with 2% pentobarbital sodium, apical blood was collected into a 10 mL anticoagulant tube and temporarily stored at 4 °C. Subsequently, hippocampal tissue was harvested, treated with concentrated nitric acid, and heated to 100 °C for digestion. Both the blood and hippocampal tissue samples were diluted using a proprietary diluent for analysis with an AAS. The concentration of lead in the samples was determined, and the measured lead levels were subjected to statistical analysis following quality standardization.

### 2.6. Brain Stereolocalization

Brain stereolocalization was performed on mice anesthetized with 2% pentobarbital sodium. The animals’ heads were immobilized using a stereotaxic apparatus, with the hippocampal CA1 region designated as the target site. The coordinates for the injection were determined as follows: anteroposterior (AP) −1.9 mm from the Bregma point, mediolateral (ML) ±1.5 mm relative to the midline, and dorsoventral (DV) −2.0 mm from the skull surface. These coordinates facilitated the precise navigation of the syringe through the brain stereoscope, allowing for the accurate administration of the high-expression Mitofilin lentiviral vector into the specified brain region. Following the injection, the mice were returned to their respective holding environments.

### 2.7. Morris Water Maze

Submerge the hidden platform 1 cm underwater in Quadrant 1 and maintain the water at 25 °C. During days 1–5 of the positioning experiment, place mice at the farthest point from the platform facing the pool wall, then remove and dry them to prevent stress. Record the incubation period daily. If a mouse does not find the platform within 90 s, guide it to stay on the platform for 10 s. On day 6, remove the platform for space exploration experiments, and release the mice facing the pool wall to track their movement.

### 2.8. Open Field Test

The mice were positioned at the center of a 40 cm × 40 cm open chamber to acclimate to the open field environment for a duration of two hours. At the onset of the experiment, the mice were again placed at the center of the chamber’s base, and their movement trajectories were recorded using the Supermaze video system. The system documented the total distance traveled, the duration spent in the central area within a five-minute interval, as well as the overall distance covered by the mice and their average speed.

### 2.9. Novel Object Recognition Test

Prior to the commencement of the experiment, the mice were situated in an open laboratory environment measuring 40 cm by 40 cm for a duration of one hour, during which they were maintained in a state of tranquility. The formal experimental procedure was bifurcated into a training phase and a testing phase. During the training phase, two identical objects were positioned at diagonal locations within the experimental enclosure. In the testing phase, novel objects of varying sizes and colors were introduced at arbitrary diagonal positions within the experimental enclosure.

### 2.10. Cellular Immunofluorescence

Polylysine-coated slides were incubated at 37 °C with 5% CO_2_ and saturated humidity for 4 h. Subsequently, the cell slides were washed three times with phosphate-buffered saline (PBS) and fixed with 4% paraformaldehyde for 15 min. Following fixation, the cells were permeabilized with a PBS solution containing 0.3% Triton X-100 for 35 min. Finally, the cells were blocked with a PBS solution containing 5% fetal bovine serum albumin (BSA) at room temperature for 45 min. The initial incubation involved the anti-mouse anti-DNA antibody at a dilution of 1:50 and the rabbit anti-TOM20 antibody at a dilution of 1:250, maintained overnight at 4 °C. Subsequently, the antibodies were subjected to three washes with PBS. Following this, the FITC-labeled goat anti-mouse secondary antibody and the CY3-labeled goat anti-rabbit secondary antibody, both at a dilution of 1:500, were incubated at room temperature for one hour. This was followed by another set of three washes with PBS. The specimens were then mounted using anti-fade mounting medium containing DAPI and examined using an ultra-high-resolution live cell imaging system.

### 2.11. Transmission Electron Microscope

The cells were harvested through centrifugation at ambient temperature, followed by the removal of the supernatant. Subsequently, the cells were fixed at 4 °C for 24 h using 2.5% glutaraldehyde. After removing the fixative, the sample was rinsed with PBS and then fixed again at 4 °C with 1% osmium tetroxide for 1 h. Following this, the sample was washed with PBS and subjected to a graded dehydration series using anhydrous ethanol, progressing through concentrations of 30%, 50%, 70%, 80%, 90%, 95%, and finally, 100%. Each dewatering step lasted for 15 min, with the final dewatering step extending to 25 min using 100% acetone. The samples were then immersed in the embedding medium and incubated at room temperature for 12 h. Following incubation, the laminated sheets were placed in a constant temperature drying oven set at 60 °C for 48 h. Subsequently, the embedded samples were sectioned into slices with a thickness of 60–80 nm using an ultra-thin microslicer. The slices were then impregnated with 1% uranyl acetate for 10 min. After rinsing with PBS, the images were examined using transmission electron microscopy.

### 2.12. Golgi Staining *[27]*

The entire brains of mice were extracted from the skull post-mortem and immersed in 5 mL of Golgi mixture AB solution (A:B = 1:1), subsequently stored at room temperature for two weeks, and shielded from light exposure. Following this period, coronal sections of the brain, each with a thickness of 100 μm, were prepared. The brain slices were rinsed twice with distilled water for 4 min per rinse. Subsequently, the slices were immersed in a Golgi mixture (D:E:ddH_2_O = 1:1:2) for staining over a 10 min duration. After staining, the slices were rinsed with distilled water and then subjected to a dehydration process using ethanol solutions at concentrations of 50%, 75%, and 95%. Subsequently, the samples were dehydrated four times using anhydrous ethanol, each for 5 min. Following this, xylene transnatol tablets were applied three times, each for 5 min. After sealing the film, the samples were dried away from light, and the number and morphology of dendritic spines were observed under a microscope.

### 2.13. Reverse Transcription-Polymerase Chain Reaction (RT-qPCR)

A volume of 200 µL of Trizol lysate was introduced into the cell culture dish, subsequently collected into a 1.5 mL enzyme-free centrifuge tube, and subjected to centrifugation. The supernatant was then discarded. Trichloromethane was added to the solution, followed by vortexing with 500 µL, and the mixture was centrifuged at 4 °C. The resulting supernatant was transferred to a new 1.5 mL centrifuge tube, where an equal volume of isopropyl alcohol was added. This mixture was vortexed and centrifuged again at 4 °C. Following quantification, RNA was reverse-transcribed into complementary DNA (cDNA), and real-time fluorescence

Quantitative PCR was conducted using the QuantStudio 7 system. Data analysis was performed utilizing the 2^−ΔΔCt^ method. The upstream primer of Immt is F–Immt: AGGGAGACACTCCAGCTTCA, and R-Immt: CTTGCTTTTCCTGTTGCGCT.

### 2.14. Western Blot

Protein quantification in the cell samples was performed utilizing the BCA protein quantification kit. The samples were subjected to boiling in a metal bath at 100 °C for 10 min. Subsequently, gel electrophoresis was conducted, initially at 80V and then at 120 V for 30 min, until the bands corresponding to the target protein were observed. Following electrophoresis, the process was halted, and the samples were placed in a 250 mA ice bath for 95 min. Mitofilin, NLRP3, IL-1β, and β-tubulin antibodies (dilution 1:1000) derived from rabbits and β-actin antibodies (1:500) derived from mice were incubated at room temperature with 5% BSA for 1.5 h. Following an 18 h incubation period, the PVDF membrane was extracted and subjected to three washes with TBST, each lasting 10 min. Subsequently, the membrane was incubated with a rabbit secondary antibody at a dilution of 1:1000 at room temperature for 1.5 h. An ECL supersensitive luminescent solution was prepared under light-avoiding conditions for the development process. The absorbance values of the bands were then analyzed using image analysis software.

### 2.15. MDA and ROS Detection

A serum-free culture medium was employed to dilute the fluorescence probe (DCFH-DA: 10 mM) at a 1:1000 ratio under light-avoiding conditions to prepare Reagent 1. Similarly, a serum-free culture medium was utilized to dilute 10 mM TBHP to formulate the positive control working solution, referred to as Reagent 2. Cells were harvested via centrifugation in a 1.5 mL centrifuge tube. Following the washing procedure, the cells were resuspended in Reagent 1. Subsequently, the cells were incubated at 37 °C for 45 min. Upon completion of the incubation period, the cells were collected, washed, and resuspended. The excitation wave of 500 nm and the emission wavelength of 525 nm were set, and the fluorescence value was detected in the enzyme marker. The TBA storage solution and MDA working solution were prepared in accordance with the instructions provided in the kit. Cell lysis was performed using a strong lysate, and the supernatant was collected by centrifugation into a new 1.5 mL centrifuge tube. The protein concentration was determined using the BCA method. For each experimental group, 0.1 mL samples were taken, and 0.2 mL of the MDA working solution was added. The samples were vortex-mixed and incubated in a boiling water bath for 15 min, followed by cooling to room temperature. Subsequently, 200 µL of the sample was transferred to 96-well plates, and the absorbance at 532 nm was measured using an ELISA reader.

### 2.16. Statistical Analysis

All experiments in this study were carried out independently and were repeated at least 3 times. Graphpad Prism 8.0.2 software was used for statistical analysis, and the data were expressed as x ± s. A t test was used for the two groups of samples, and two-way analysis of variance (two-way ANOVA) was used for the comparison of multiple groups. *p* < 0.05 indicated that the difference was statistically significant.

## 3. Results

### 3.1. The Chronic Lead Exposure Impaired the Spatial Memory and Learning of Mice

AAS analysis revealed that, in comparison to the control group, mice subjected to chronic lead exposure exhibited a significant increase in both blood and brain lead concentrations (Figure 1A). Subsequently, the impact of chronic lead exposure on the learning and memory capabilities of the mice was further investigated. The findings of the water maze experiment indicated that there was no statistically significant difference in the ratio of swimming speed to body weight between the chronic lead exposure group and the control group, suggesting that chronic lead exposure does not impact the exercise capacity of mice (Figure 1B,C). However, in comparison to the control group, the latency period for mice in the chronic lead exposure group progressively decreased from day 1 to day 5 (Figure 1D,E). Furthermore, there was a significant reduction in the total distance traveled, duration, time, and frequency of platform crossings in the target quadrant for the chronic lead exposure group (Figure 1F–I). The findings from the novel object recognition experiment indicated that, relative to the control group, the cognitive index and the frequency of novel object exploration in mice subjected to chronic lead exposure were significantly reduced (Figure 1J–L). Conversely, results from the open field test demonstrated no statistically significant differences between the chronic lead exposure group and the control group in terms of total distance traveled, average velocity, distance covered in the central area, and duration spent in the open field (Figure 1M–Q). These results suggest that chronic lead exposure does not have a significant impact on anxiety-related behaviors. According to these results, chronic lead exposure can impair mice’s spatial cognition, learning, and memory abilities.

### 3.2. The Effects of Chronic Lead Exposure on Neuronal Dendritic Spine Plasticity and Mitochondrial Structure and Function

The CA1 region of the hippocampus serves as a crucial structural component for learning and memory in mice, comprising a network of neural circuits that are integral to higher-order brain functions, including learning, memory, and the transformation of movement scenes [28,29]. The plasticity of neuronal dendritic spines, a vital mechanism for sustaining neuronal function, is intimately associated with the processes of learning and memory [30,31,32]. The results of Golgi staining indicated that, in comparison to the control group, the chronic lead exposure group exhibited a significant reduction in the density of dendritic spines within the hippocampal CA1 region (Figure 2A,B). Additionally, there was a notable shortening in the length of both basal and apical dendritic spines (Figure 2C). Specifically, the density of dendritic spines in the hippocampal CA1 region was significantly decreased in the mushroom subtype, while an overall increase in stubby dendritic spine density was observed (Figure 2D,E). Given that mitochondria are highly energy-demanding organelles, their structural integrity is critically linked to neuronal function [33]. Recent research indicates that mitochondrial homeostasis plays a crucial role in influencing the plasticity of neuronal dendritic spines [34,35]. Transmission electron microscopy analysis revealed that, in comparison to the control group, neuronal mitochondria in the hippocampal CA1 region of mice subjected to chronic lead exposure exhibited swelling and matrix vacuolation as a result of ridge fracture (Figure 2F). Furthermore, the integrity of the mitochondrial structure is closely associated with the level of oxidative stress [36,37]. The findings from the MDA and ROS detection assays indicated a significant elevation in MDA and ROS levels in HT22 cells and in the hippocampus subjected to chronic lead exposure, in comparison to the control group (Figure 2G–K). Mitofilin, a crucial protein located in the inner mitochondrial membrane, is essential for preserving mitochondrial structural integrity [38]. Western blot analyses revealed that the levels of the Mitofilin protein were markedly reduced in the HT22 cells and in the hippocampus tissue of the chronic lead exposure group relative to the control group (Figure 2L–O).

RT-qPCR analyses revealed that the mRNA expression levels of Immt were markedly reduced in the HT22 cells of the chronic lead exposure group relative to the control group (Figure 2P).These results suggest that chronic lead exposure may affect mitochondrial structural integrity by reducing the expression level of Mitofilin, thereby mediating the plasticity damage of neuronal dendritic spines.

### 3.3. Overexpression of Mitofilin Improves Learning and Memory in Mice by Affecting the Expression of Mitofilin After Chronic Lead Exposure

Our research team engineered the AAV-SYN- tdTomato-P2A-Mitofilin virus to elevate Mitofilin expression levels specifically in neurons within the CA1 region of the hippocampus, utilizing the stereotaxic injection technique (Figure 3A,B). Results from the water maze experiment indicated that, in comparison to the chronic lead exposure group, mice with both chronic lead exposure and Mitofilin overexpression exhibited a progressive reduction in latency time from day 1 to day 5 (Figure 3C,D). And the total distance, duration, time taken to cross the platform, and frequency of platform crossings in the target quadrant were significantly increased (Figure 3E–H). Subsequent results from the novel object recognition experiment indicated that, in comparison to mice subjected solely to chronic lead exposure, those in the chronic lead exposure group with Mitofilin overexpression exhibited a significant increase in cognitive indices and the frequency of exploration of novel objects (Figure 3I–K). Western blot analyses revealed that levels of the Mitofilin protein were markedly increased in the HT22 cells’ overexpression in the chronic lead exposure group relative to the chronic lead exposure group (Figure 2L,M). Western blot analyses revealed that the levels of Mitofilin protein were markedly increased in the hippocampus tissue overexpression in the chronic lead exposure group relative to the chronic lead exposure group (Figure 2N,O). These results suggest that overexpression of Mitofilin can ameliorate the decreased learning and memory ability of mice caused by chronic lead exposure.

### 3.4. A Mitochondrial Structural Integrity Improvement by Overexpressing Mitofilin Protects Neuronal Dendritic Spines from Lead-Induced Plasticity Damage

The results of Golgi staining indicated that, in comparison to the chronic lead exposure group, the group subjected to chronic lead exposure combined with Mitofilin overexpression exhibited a significant increase in the density of neuronal dendritic spines in the hippocampal CA1 region. Additionally, there was a notable increase in the length of both basal and apical dendritic spines. Furthermore, the density of mushroom-type neuronal dendritic spines in the hippocampal CA1 region was significantly elevated, whereas the density of stubby-type neuronal dendritic spines in the same region was significantly reduced (Figure 4A–E). Subsequent analysis using transmission electron microscopy revealed that, relative to the chronic lead exposure group, the integrity of the mitochondrial cristae structure in neurons within the hippocampal CA1 region of mice was more preserved in the group subjected to chronic lead exposure combined with Mitofilin overexpression, with a noted reduction in mitochondrial matrix vacuolation (Figure 4F). Furthermore, in comparison to the chronic lead exposure group, the levels of ROS and MDA in HT22 cells were significantly lower in the group experiencing chronic lead exposure alongside Mitofilin overexpression (Figure 4G,H). These results suggest that overexpression of Mitofilin protects neuronal dendritic spine plasticity damage after lead exposure by improving mitochondrial structural integrity.

### 3.5. Inhibition of the Release of mtDNA by Mitofilin Overexpression Diminishes the Resultant Damage to Neurons

Research has demonstrated a strong correlation between the release of mtDNA and the structural integrity of mitochondria. Serving as a pivotal hub for downstream inflammatory signaling, mtDNA is released in substantial amounts to activate the Nlrp3 inflammasome and induce pathological conditions [39,40,41]. Cellular immunofluorescence analyses revealed a significant increase in mtDNA release in HT22 cells subjected to chronic lead exposure compared to the control group. Conversely, overexpression of Mitofilin resulted in a marked reduction in mitochondrial mtDNA release relative to the chronic lead exposure group (Figure 5A,B). Nlrp3 serves as a critical inflammatory signaling molecule downstream of mtDNA, with its activation status being intricately associated with neurotoxicity following chronic lead exposure [42,43]. Western blot analyses revealed a significant upregulation of Nlrp3 and IL-1β protein levels in HT22 cells subjected to chronic lead exposure, in comparison to the control group. Conversely, the overexpression of Mitofilin resulted in a marked reduction in the levels of Nlrp3 and IL-1β proteins relative to the chronic lead exposure group (Figure 5C–E). Western blot analyses revealed a significant upregulation of Nlrp3 and IL-1β protein levels in the hippocampus tissue subjected to chronic lead exposure, in comparison to the control group. Conversely, the overexpression of Mitofilin resulted in a marked reduction in the levels of Nlrp3 and IL-1β proteins relative to the chronic lead exposure group (Figure 5F–H).

These results suggest that Mitofilin can alleviate neuronal inflammatory damage by inhibiting the release of mtDNA, and thus play an important role in protecting the plasticity of neuronal dendritic spines.

## 4. Discussion

Studies have shown that lead, mercury, and cadmium persist in the brain for extended periods, resulting in neurotoxic effects [44]. Lead enters the brain through the blood–brain barrier (BBB), and early exposure to lead can significantly affect intellectual development during pregnancy and infancy [45,46]. Some studies confirm that chronic lead exposure can lead to reduced learning and memory abilities, impaired fine motor skills and spatial cognition, and changes in executive function [1]. Subsequent research has demonstrated that lead exposure can directly impair aging neurons through mechanisms including apoptosis, mitochondrial oxidative damage, aberrant reactive oxygen species (ROS) production, disruption of neurotransmitter storage and release, enzyme inactivation, and abnormal oxidative stress, ultimately resulting in neurodegeneration [47,48]. Epidemiological research has indicated that significant lead accumulation in the nervous system may contribute to the pathogenesis of neurodegenerative disorders, such as Alzheimer’s disease (AD), Parkinson’s disease (PD), and amyotrophic lateral sclerosis [49]. This study effectively established an in vivo model of chronic lead exposure based on previous studies [26], further confirming that chronic lead exposure can impair learning and memory ability in mice.

Neuronal dendritic spine plasticity facilitates learning and memory processes through dynamic alterations in the size, shape, and quantity of dendritic spines. In healthy mice, these cognitive functions are linked to intricate interactions among neurons within the hippocampus, which rely on dendritic spine plasticity [50]. Research has demonstrated that prolonged lead exposure disrupts synaptic plasticity, thereby detrimentally impacting synaptic function and neuronal morphology in the CA1 region of the hippocampus, ultimately leading to compromised higher-order brain functions [51]. In this study, Golgi staining was used to detect the plasticity of dendritic spines. It was found that the density of dendritic spines in the hippocampal CA1 region of mice was significantly reduced after chronic lead exposure, and the dendritic spines at the base and apex were significantly shortened, which was consistent with some previous studies. These results suggest that chronic lead exposure can cause plasticity damage of neuronal dendritic spines in the CA1 region of the hippocampus in mice.

The programmed changes in mitochondrial morphology and metabolic function are closely related to the normal execution of neuronal function [52]. Studies have shown that the dynamic changes in mitochondrial morphology are significantly related to the typical characteristics of the synaptic plasticity of neurons [53]. Several research groups have demonstrated that chronic lead exposure not only significantly impairs mitochondrial function and induces oxidative stress within mitochondria but also disrupts the homeostasis of the mitochondrial cristae structure, leading to the release of substantial amounts of mitochondrial DNA (mtDNA). This disruption results in neuronal inflammatory damage and mediates pathological processes [54,55]. In the present study, an in vitro model of lead exposure using HT22 cells was established. Findings indicated that after 24 h of chronic lead exposure, there was a marked increase in mitochondrial oxidative stress levels, accompanied by mitochondrial swelling and damage to the cristae structure. Furthermore, lead exposure induced alterations in mitochondrial morphology, including changes in mitochondrial fusion, fission, and cristae remodeling.

The study revealed that Mitofilin plays a dual role in enhancing neuronal mitochondrial quality via the mitochondrial autophagy pathway and in regulating mitochondrial morphology homeostasis, thereby mediating improvements in mitochondrial function and the redox state [56,57]. Our findings indicated a significant reduction in Mitofilin expression in HT22 cells 24 h post-lead exposure, implying that lead exposure may compromise the structural integrity of neuronal mitochondria by down-regulating Mitofilin expression. The disruption of the mitochondrial structure can facilitate the release of mtDNA into the cytoplasm, triggering inflammatory pathways and potentially leading to various neurological disorders [39]. Our research demonstrated that Mitofilin overexpression significantly reduced neuronal mtDNA release, decreased oxidative stress levels, improved mitochondrial structural integrity in the hippocampal CA1 region, increased neuronal dendritic spine density, and, ultimately, partially restored learning and memory functions.

This research focuses on the role of the Mitofilin–mtDNA axis in the neuronal dendritic spine morphology and cognitive functions related to learning and memory. While the study has elucidated the function of Mitofilin, the precise mechanisms by which the Mitofilin protein within the mitochondrial inner membrane regulates mtDNA release remain to be fully understood. Recent reports suggest that PINK1 and IGF2 are involved in mitochondrial quality control and redox status by modulating the binding of Mitofilin to the mitochondrial inner membrane, providing a reference point for exploring upstream regulatory mechanisms in the context of chronic lead exposure models in this study [56,57]. In addition, this study confirmed that the mitochondrial intimal protein Mitofilin is closely related to the plasticity of neuronal dendritic ridges, but it still lacks functional indicators such as neuronal electrophysiology to verify it.

## 5. Conclusions

Our study demonstrated that the Mitofilin protein is a crucial molecule influencing the damage to neuronal dendritic spine plasticity following chronic lead exposure. We identified that the overexpression of the Mitofilin protein exerts a protective effect on the structural integrity of neuronal mitochondria and facilitates the recovery of mitochondrial function. The research confirmed that the down-regulation of the Mitofilin protein leads to increased production of malondialdehyde (MDA) and reactive oxygen species (ROS), thereby inducing the release of mitochondrial DNA (mtDNA) and elevating the expression of downstream inflammatory factors. Furthermore, the study elucidated the significant regulatory role of the Mitofilin–mtDNA axis in hippocampal synaptic plasticity damage and cognitive deficits induced by chronic lead exposure. In conclusion, the potential role of Mitofilin in preventing and treating neurotoxicity caused by chronic lead exposure merits further investigation and attention.

## Figures and Tables

**Figure 1 biomolecules-15-00272-f001:**
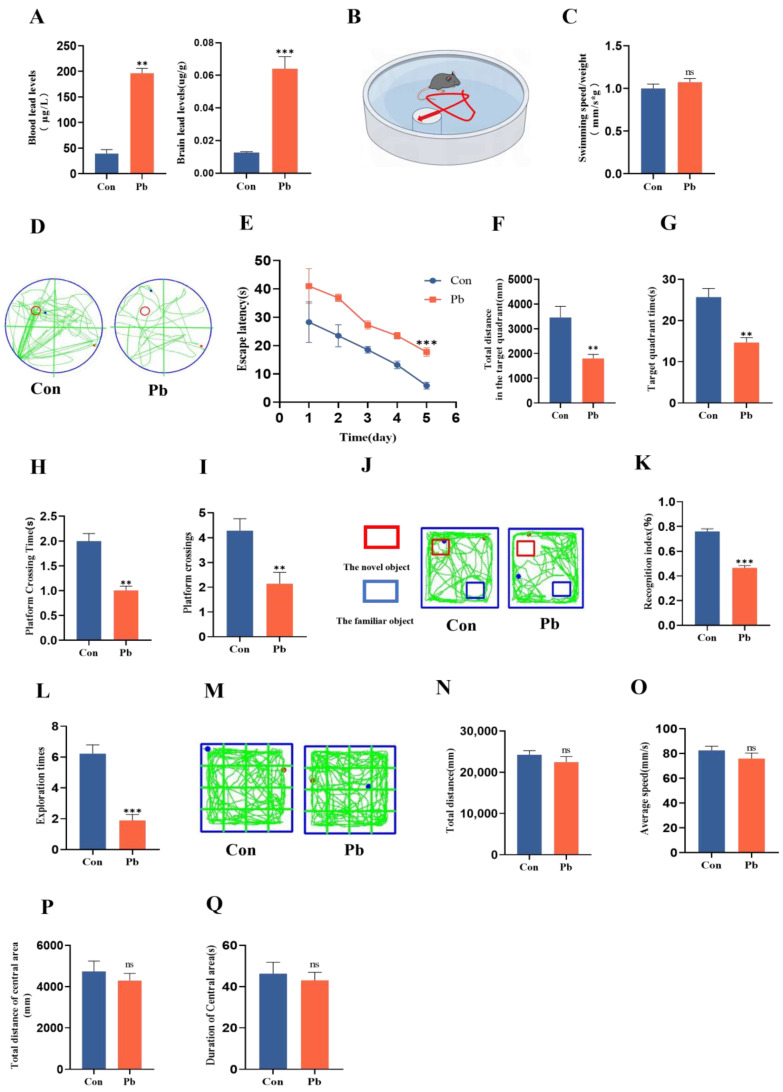
Spatial learning and memory ability of mice damaged by chronic lead exposure. (**A**) The concentrations of blood and brain lead in mice following chronic lead exposure were measured using atomic absorption spectroscopy (AAS). (**B**) A representative model of the water maze experiment in mice subjected to chronic lead exposure. (**C**) The ratio of swimming speed to body weight in mice after chronic lead exposure was subjected to statistical analysis. (**D**) The water maze test trajectory of mice following chronic lead exposure(the green line represents the animal’s movement track, and the red circle is the platform location of the animal water maze experiment). (**E**) Statistical analysis of the escape latency period in mice after chronic lead exposure. (**F**) Statistical analysis of the total distance traveled in the target quadrant by mice following chronic lead exposure. (**G**) Statistical analysis of the duration spent in the target quadrant following chronic lead exposure. (**H**) Statistical analysis of the penetration time subsequent to chronic lead exposure. (**I**) Statistical analysis of the frequency of platform crossings after chronic lead exposure. (**J**) Experimental trajectory of novel object recognition in mice following chronic lead exposure(the green line represents the animal’s movement track). (**K**) Statistical analysis of the novel object cognition index subsequent to chronic lead exposure. (**L**) Statistical analysis of the frequency of novel object exploration following chronic lead exposure. (**M**) Tracking chart of mice in the open field experiment following chronic lead exposure(the green line represents the animal’s movement track). (**N**) Statistical analysis of the total distance covered in the open field after chronic lead exposure. (**O**) Statistical analysis of the average velocity in the open field following chronic lead exposure. (**P**) Statistical analysis of the total distance traveled in the central area of the open field after chronic lead exposure. (**Q**) Statistical analysis of the duration spent in the open field following chronic lead exposure. Data are presented as the mean ± SEM (n = 8). ** *p* < 0.01 vs. Con; *** *p* < 0.001 vs. Con.ns, not significant.

**Figure 2 biomolecules-15-00272-f002:**
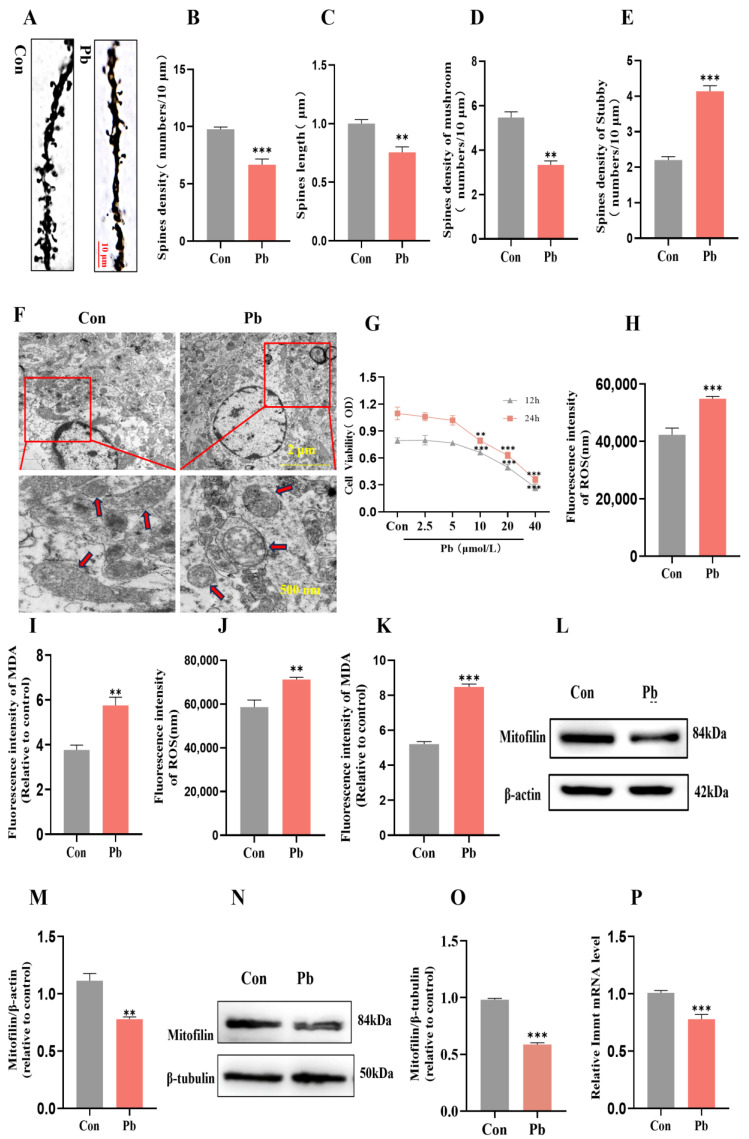
The effects of chronic lead exposure on neuronal dendritic spine plasticity and mitochondrial structure and function. (**A**) Golgi staining was employed to examine alterations in neuronal dendritic spines within the CA1 region of the hippocampus in mice subjected to chronic lead exposure. (**B**) The density of neuronal dendritic spines in the hippocampal CA1 region was subjected to statistical analysis. (**C**) The length of dendritic spines in neurons of the hippocampal CA1 region was statistically analyzed. (**D**) The density of mushroom-type neuronal dendritic spines in the CA1 region of the hippocampus was statistically analyzed. (**E**) The density of stubby dendritic spines in neurons of the hippocampal CA1 region was statistically analyzed. (**F**) The mitochondrial structure of neurons in the CA1 region of the hippocampus was examined using transmission electron microscopy following chronic lead exposure (The red arrows indicate mitochondria; The red square indicates local magnification). (**G**) The effect of lead exposure on HT22 cell viability was determined by MTT. (**H**) Alterations in reactive oxygen species (ROS) levels in HT22 cells were assessed after chronic lead exposure. (**I**) Modifications in malondialdehyde (MDA) levels in HT22 cells were evaluated following chronic lead exposure. (**J**) Alterations in reactive oxygen species (ROS) levels in the hippocampus were assessed after chronic lead exposure. (**K**) Modifications in malondialdehyde (MDA) levels in the hippocampus were evaluated following chronic lead exposure. (**L**) Western blot analysis was conducted to determine the expression of Mitofilin in HT22 cells after chronic lead exposure. (**M**) Gray-scale analysis was performed to quantify the expression level of Mitofilin in HT22 cells following chronic lead exposure. (**N**) Western blot analysis was conducted to determine the expression of Mitofilin in the hippocampus tissue after chronic lead exposure. (**O**) Gray-scale analysis was performed to quantify the expression level of Mitofilin in the hippocampus tissue following chronic lead exposure. (**P**) RT-qPCR was employed to measure the expression level of Immt mRNA in HT22 cells after chronic lead exposure. Data are presented as the mean ± SEM (n = 5). ** *p* < 0.01 vs. Con; *** *p* < 0.001 vs. Con. Western blot original images are in the Appendix A.

**Figure 3 biomolecules-15-00272-f003:**
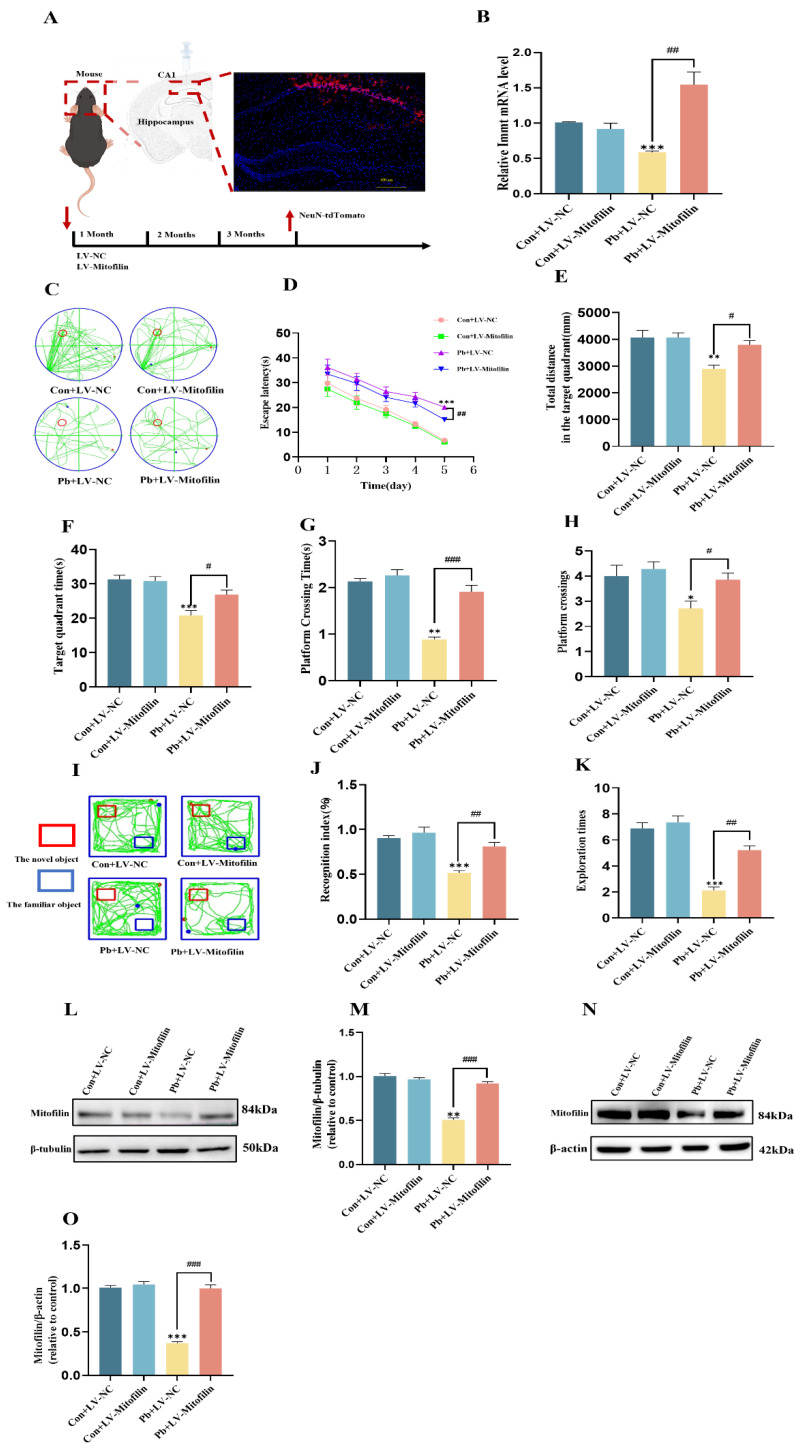
Overexpression of Mitofilin improves learning and memory in mice by affecting the expression of Mitofilin after chronic lead exposure. (**A**) Stereotaxic injection of AAV-SYN- tdTomato-P2A-Mitofilin virus into the mouse brain to create a model map. (**B**) Statistical analysis of mRNA expression levels in the hippocampus of mice following Immt expression, as detected by RT-qPCR. (**C**) Assessment of the water maze test trajectories in mice post-Mitofilin overexpression(The green line represents the animal trajectory, and the red circle represents the location of the water maze experiment platform.). (**D**) Statistical analysis of the escape latency period following Mitofilin overexpression. (**E**) Statistical evaluation of the total distance covered in the target quadrant after Mitofilin overexpression. (**F**) Statistical analysis of the duration spent in the target quadrant following Mitofilin overexpression. (**G**) The duration of Mitofilin overexpression was subjected to statistical analysis. (**H**) Statistical analysis was conducted on the frequency of platform crossings following Mitofilin overexpression. (**I**) An experimental track map was generated to assess the interaction with new objects in mice post-Mitofilin overexpression(The red square represents the new object position). (**J**) The cognitive index related to new objects following Mitofilin overexpression underwent statistical evaluation. (**K**) The frequency of new object exploration subsequent to Mitofilin overexpression was analyzed statistically. (**L**) Western blot analysis was conducted to determine the expression of overexpressing Mitofilin in HT22 cells after chronic lead exposure. (**M**) Gray-scale analysis was performed to quantify the expression level of overexpressing Mitofilin in HT22 cells following chronic lead exposure. (**N**) Western blot analysis was conducted to determine the expression of overexpressing Mitofilin in hippocampus tissue after chronic lead exposure. (**O**) Gray-scale analysis was performed to quantify the expression level of overexpressing Mitofilin in hippocampus tissue following chronic lead exposure. Data are presented as the mean ± SEM (n = 8). * *p* < 0.05 vs. Con+LV−NC; ** *p* < 0.01 vs. Con+LV−NC; *** *p* < 0.001 vs. Con+LV−NC; ^#^ *p* < 0.05 vs. Pb+LV−NC; ^##^ *p* < 0.01 vs. Pb+LV−NC; ^###^ *p* < 0.001 vs. Pb+LV−NC. Western blot original images are in the Appendix A.

**Figure 4 biomolecules-15-00272-f004:**
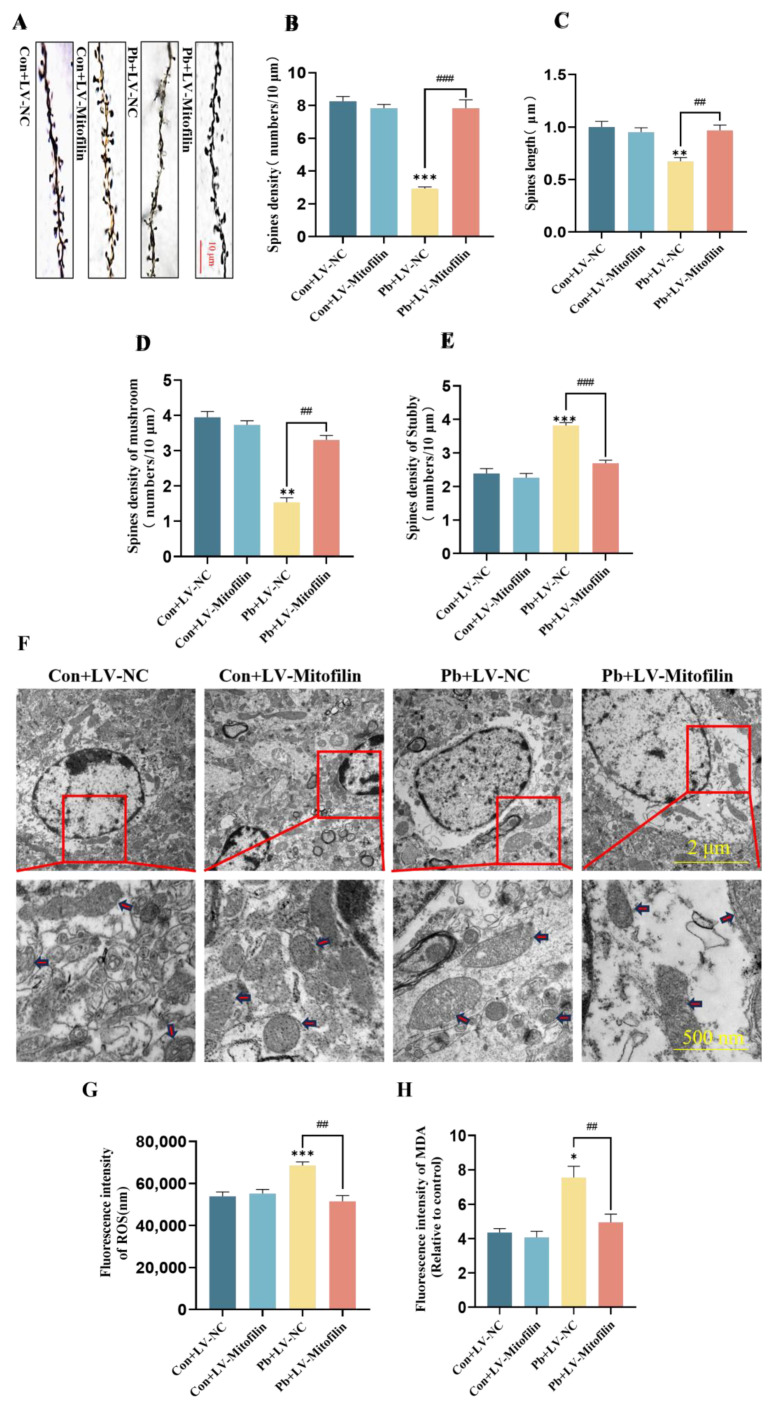
Mitofilin overexpression protects neuronal dendritic spine plasticity damage after lead exposure by improving mitochondrial structural integrity. (**A**) Alterations in neuronal dendritic spines were examined using Golgi staining. (**B**) A statistical analysis was performed on the dendritic spine density of neurons within the CA1 region of the hippocampus. (**C**) A statistical analysis was conducted on the length of neuronal dendritic spines in the CA1 region of the hippocampus. (**D**) A statistical analysis was carried out on the density of mushroom-shaped dendritic spines in the CA1 region of the hippocampus. (**E**) A statistical analysis was conducted on the density of stubby dendritic spines in the CA1 region of the hippocampus following the overexpression of Mitofilin. (**F**) The mitochondrial architecture of neurons in the CA1 region of the hippocampus was examined using transmission electron microscopy subsequent to Mitofilin overexpression (The red arrows indicate mitochondria; The red square indicates local magnification). (**G**) Alterations in reactive oxygen species (ROS) levels in HT22 cells were assessed following Mitofilin overexpression. (H) Modifications in malondialdehyde (MDA) levels in HT22 cells were evaluated after the overexpression of Mitofilin. Data are presented as the mean ± SEM (n = 4). * *p* < 0.05 vs. Con; ** *p* < 0.01 vs. Con; *** *p* < 0.001 vs. Con; ^##^ *p* < 0.01 vs. Pb+LV−NC; ^###^ *p* < 0.001 vs. Pb+LV−NC.

**Figure 5 biomolecules-15-00272-f005:**
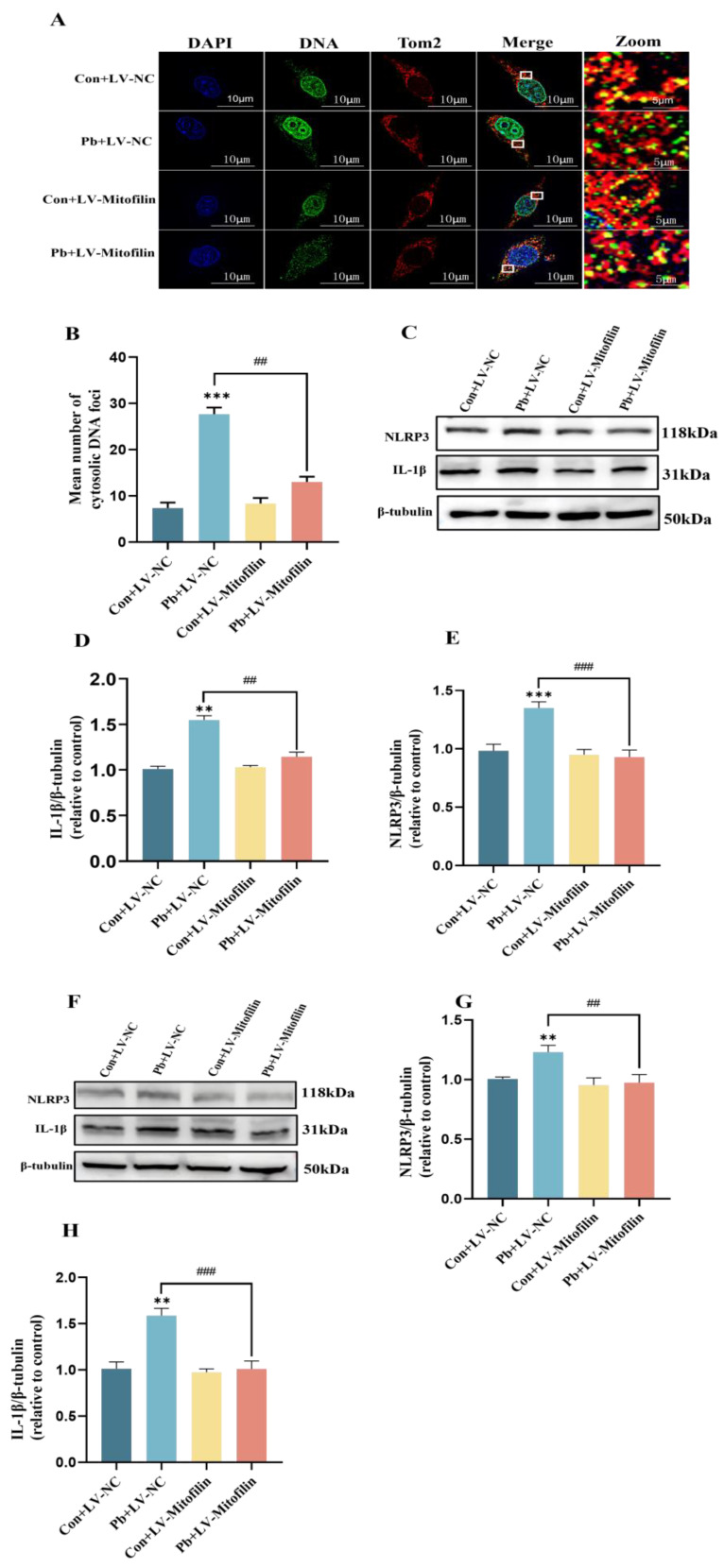
Inhibition of the release of mtDNA by mitofilin overexpression diminishes the resultant damage to neurons. (**A**) The release of mtDNA from neurons following the overexpression of Mitofilin was examined using cellular immunofluorescence staining. (**B**) Quantitative analyses were conducted to assess the alterations in mtDNA release in neurons post-Mitofilin overexpression. (**C**) The expression levels of Nlrp3 and IL-1β in HT22 cells subsequent to Mitofilin overexpression were determined via Western blot analysis. (**D**,**E**) The expression level of IL-1β in HT22 cells was evaluated through gray-scale analysis, as was the expression level of Nlrp3. (**F**) The expression levels of Nlrp3 and IL-1β in the hippocampus tissue subsequent to Mitofilin overexpression were determined via Western blot analysis. (**G**,**H**) The expression level of IL-1β in the hippocampus tissue was evaluated through gray-scale analysis, as was the expression level of Nlrp3. Data are presented as the mean ± SEM (n = 4). ** *p* < 0.01 vs. Con; *** *p* < 0.001 vs. Con; ^##^ *p* < 0.01 vs. Pb+LV−NC; ^###^ *p* < 0.001 vs. Pb+LV−NC. Western blot original images are in the Appendix A.

## Data Availability

The data generated in the present study may be requested from the corresponding author.

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
