# Peer review of "Mitofilin–mtDNA Axis Mediates Chronic Lead Exposure-Induced Synaptic Plasticity Impairment of Hippocampal and Cognitive Deficits"

_biomolecules, 2025, doi:10.3390/biom15020272_

Round 1

Reviewer 1 Report

Comments and Suggestions for Authors

I appreciate the opportunity to review the manuscript:  I appreciate the opportunity to review the manuscript:  The mitochondrial intima protein Mitofilin mediates synaptic 2 plasticity damage after chronic lead exposure” LIHONG SU et.al.

In my opinion, the manuscript is interesting, but requires additions and corrections.

Recommendation: Major revision

General comments:

Title: I would suggest changing the title: „The mitochondrial intima…”, it is incomprehensible

Introduction Line 65 and 65 In my opinion, a completely unnecessary sentence: „Structurally, they possess both inner and outer membranes, with the inner membrane exhibiting invaginations known as cristae”

“HT22 cells in lead exposure group (Pb) were cultured with 5µmol/L lead acetate medium for 24 h”. Please justify the dose of Pb used in the experiment.

“There were 8 rats in each group, which were control group (Con), Pb expo- 136 sure group (Pb), control + blank treatment group (Con+LV-NC), Pb exposure + blank treatment group (Pb+LV-NC), control + over expressing Mitofilin group (Con+LV-Mitofilin), Pb exposure + over expressing Mitofilin group (Pb+LV-Mitofilin)”.  Please explain what the abbreviations used mean LV-NC, LF, NC mean?

Please explain how is it possible that mice drank deionized water, why couldn't they drink tap water, did they not have deficiencies of other elements. In addition, such water is very unpalatable, were the mice not dehydrated? How much water did each of the groups drink daily? Please explain.

Water containing 100 ppm Pb - please explain why such a high dose of Pb was used?

The authors of the paper supplemented the water with Pb once a week (“once a week at a fixed time. To ensure that the mice drink enough water”). It is known that lead acetate solution precipitates, how did the authors deal with this problem? Please explain.

The level of lead in the blood of the animals was very high - it amounted to as much as 200ug/L (20ug/dL) why was such a high dose decided? - already 3.5 ug gives neurotoxic symptoms

Fig. 2. Graph G is incomprehensible, why doses of 5, 10, 20, 40 ug/L were tested, since the concentration of Pb in the animals' blood was 200 ug/L

Graf F- at such magnification no symptoms are visible described structures, they should be shown in better magnification and the changes in mitochondrial morphology should be precisely indicated and described. The same applies to Fig 4 F

Discussion 503 Line - information that Pb causes Wilson's disease is not supported by any citations (it is known so far that it concerns Cu poisoning, not Pb) How will the authors explain this?

 The discussion is the weakest part of the paper. It is more like an Introduction than a Discussion. The authors practically did not relate their results to the achievements of other authors. There have already been at least several papers on mitochondrial morphology disorders (in various models of Pb neurotoxicity) under the influence of Pb and it would be appropriate to address them.

Moreover the authors of the paper should first of all explain why they used such high doses of Pb; it has long been known that even 10 times lower doses are toxic.

Author Response

请参阅附件。

Reviewer 2 Report

Comments and Suggestions for Authors

The study presented in this manuscript tried to investigate the role and underlying mechanisms of Mitofilin in synaptic plasticity damage induced by chronic lead exposure. Although the obtained results are interesting and contribute to the comprehensive understanding of lead neurotoxicity mechanism, the manuscript requires further improvements. Some sentences are completely copy paste from the used literature. Authors should rewritten the manuscript in order to paraphrase the facts form the literature. Authors should justify the novelty of obtained findings. Please pay attention on specific comments given below:

-Title should be changed in order to better reflect the aim, applied methodology and obtained results

-Abstract: “heavy metal lead pollution” should be changed. Authors should use heavy metal pollution or lead pollution

-This sentence should be completely rewritten “The regulatory impact of lead exposure on neuronal dendritic spine plasticity, a crucial mechanism for neuronal adaptation, warrants further investigation”.

-Abstract should be rewritten in order to better reflect the aim, the applied methodology and the obtained results. In the current version everything is mix-matched

-Introduction section: First sentence should be deleted.

-Line 43-44: which countries? Reference is missing as well as at the end of next sentence given in lines 44-46.

-Methodology, line 131 and Line 145: What is the base for the chosen dose as well as the period of exposure? Authors should justify all of that.

-Line 283: It is well known that lead exposure negatively impaires memory and learning.  What is novel?

-Line 284-285: This should be mentioned in methodology section instead of results.

-Figure 1-5: Font size and resolution should be increased

-Line 386: LV-Mitofilin-tdTomato-AAV should be explained in details in methodology as well

-Line 498-499: The sentence should be deleted

-Line 503: Wilson's disease is a rare inherited condition that causes copper levels to build up in several organs, especially the liver, brain and eyes. Why did authors associate lead exposure with Wilson's disease?

- The mentioned facts always have to be related to the published data. Authors omitted in several paragraphs in manuscript to include references, especially in the discussion section.

-The discussion part should be improved. The obtained results should be better compared with the published data.

- The limitations of this study should be inserted before conclusion section.

-In conclusion the major findings should be mentioned. Please be more precise instead of general statement.

-References should be checked. It is not commen to insert dot after the number given in brackets

Round 2

Reviewer 2 Report

Comments and Suggestions for Authors

Although authors have made some improvements, several issues are still presented in the revised manuscript. I encourage authors to give more efforts in making this paper ssuitable for publication.  

-Abstract: First sentence should be rewritten. Authors could not use term “these animals” having in mind that in the previous sentence they only stated “In this study, both in vivo and in vitro models were subjected to chronic lead exposure.”

-Abstract: The aim of the research is missing.

-Key words should be carefully revised. “Mitochondrial” is not a proper key word.  

-This sentence should be completely rewritten “The regulatory impact of lead exposure on neuronal dendritic spine plasticity, a crucial mechanism for neuronal adaptation, warrants further investigation”.

-Line 46: in which countries? Again, authors omitted to include the names of countries.

-Introduction: The given aim of the study in lines 129-131 does not correspond well with the presented results and discussion. I encourage authors to rewrite or delete this part.

- The sentence given in lines 564-566 “Epidemiological research has indicated that significant lead accumulation in the nervous system may contribute to the pathogenesis of neurodegenerative disorders, such as Alzheimer's disease (AD), Parkinson's disease (PD), and amyotrophic lateral sclerosis [44].” should be placed after the sentence given in line 573-577 “….neurogeneration [48,49].

-Line 567: The sentence “The brain is the primary target organ for exposure to environmental heavy metals”  should be deleted.

- Line 578-579-authors should cite “previous studies”.

-The last sentence in the conclusion section should be rewritten. Please avoid the term “scholarly attention”.

Author Response

请参阅附件
